# Myeloid-Derived Suppressor Cells in Prostate Cancer: Present Knowledge and Future Perspectives

**DOI:** 10.3390/cells11010020

**Published:** 2021-12-22

**Authors:** Filippos Koinis, Anastasia Xagara, Evangelia Chantzara, Vassiliki Leontopoulou, Chrissovalantis Aidarinis, Athanasios Kotsakis

**Affiliations:** 1Department of Medical Oncology, University General Hospital of Larissa, 41221 Larissa, Thessaly, Greece; phillipkoinis@gmail.com (F.K.); valiaxantzara@gmail.com (E.C.); vasoula_leontop@yahoo.com (V.L.); valadisaidarinis@yahoo.com (C.A.); 2Laboratory of Oncology, Faculty of Medicine, School of Health Sciences, University of Thessaly, 41500 Larissa, Thessaly, Greece; xagaraa@hotmail.com

**Keywords:** MDSCs, prostate cancer, immunosuppression, immunotherapy

## Abstract

Several lines of research are being investigated to better understand mechanisms implicated in response or resistance to immune checkpoint blockade in prostate cancer (PCa). Myeloid-derived suppressor cells (MDSCs) have emerged as a major mediator of immunosuppression in the tumor microenvironment that promotes progression of various tumor types. The main mechanisms underlying MDSC-induced immunosuppression are currently being explored and strategies to enhance anti-tumor immune response via MDSC targeting are being tested. However, the role of MDSCs in PCa remains elusive. In this review, we aim to summarize and present the state-of-the-art knowledge on current methodologies to phenotypically and metabolically characterize MDSCs in PCa. We describe how these characteristics may be linked with MDSC function and may influence the clinical outcomes of patients with PCa. Finally, we briefly discuss emerging strategies being employed to therapeutically target MDSCs and potentiate the long-overdue improvement in the efficacy of immunotherapy in patients with PCa.

## 1. Introduction

The role of immunotherapy in the management of patients with prostate cancer (PCa) is controversial. PCa is the first adult solid tumor to benefit from a cancer vaccine (sipuleucel-T). Although its efficacy is only modest, this finding demonstrates the potential for immunotherapy in PCa patients. Promising preclinical [1] and clinical data [2,3] provided proof of concept for considering metastatic castration-resistant (mCRPC) patients as suitable candidates for treatment with immune checkpoint blockade (ICB). However, despite the efficacy in individual patients, phase III trials in unselected patients failed to demonstrate a survival benefit [4,5]. These results are not surprising given the absence of biomarkers to guide the selection of patients and monitor treatment efficacy or develop more effective combinatorial strategies [6].

Nonetheless, several preliminary findings suggest that the observed resistance to single-agent ICB is reversible. An integral component of these efforts is the understanding of the mechanisms usurped by the cancer to escape host immune surveillance. To this end, ICB is being combined with molecularly targeted agents [7,8,9]. Indeed, the immunosuppressive tumor microenvironment (TME) in PCa has been experimentally reversed using PARP inhibitors [7], second-generation antiandrogen blockade [10], or anti-VEGF treatment [11,12]. Despite the proposed mechanism that may account for the clinical observations, critical studies to link the experimental data with the clinical trial results have not been conducted. Thus, an emerging critical unmet need is to elucidate the mechanisms of resistance to ICB and leverage the understanding for developing predictive markers to inform the application of immunotherapy clinically.

The main premise being considered is that PCa develops potent immunosuppressive mechanisms, limiting the ability of the host immune system to detect cancer cells and control tumor growth. These include the negative regulators of T-cell function, such as CTLA-4 and PD-1 [13], and the immune camouflage (low levels of surface MHC and low mutational load) of PCa cells [14]. At a cellular level, complex interactions between innate and adaptive immune compartments, as well as the role of the endothelial cells and the vasculature, have been considered to function towards the establishment of cell-mediated immunosuppression [15]. To this end, MDSCs have been identified as key contributors to the establishment of an immunosuppressive and pro-tumorigenic TME in PCa.

## 2. MDSCs in Cancer

MDSCs were first described in the early 1970s as a cell population that naturally suppresses the activity and function of cytotoxic T cells [16,17]. Notably, this cell population was phenotypically distinct from B and T lymphocytes, NK cells and macrophages, but similar to monocytes and neutrophils [18]. Interestingly, these dedifferentiated cells could functionally inhibit anti-tumor immune activity and promote immune evasion [19]. More than three decades of research on elucidating the origin and biological activity of this population led, in 2007, to the establishment of their name [20]. We now know that MDSCs are a highly heterogeneous population of cells originating from the bone marrow with immunosuppressive activity. Two distinct MDSC subpopulations have been identified in humans: a population resembling immature granulocytes (G-MDSC) and a population with a monocytic morphology (M-MDSC) [21]. Each has discrete immunosuppressive properties including inhibition of T-cell activation [22], dendritic cell maturation [23], induction of anergy of natural killer cells [24], and promotion of a de novo expansion of Tregs [25] in different types of malignancies. In addition to their immune-suppressive properties, MDSCs in the TME can differentiate into endothelial cells, fibroblasts, tumor-associated macrophages (TAMs) [26], and osteoclasts [27] under the influence of chemokines released by tumor cells (Figure 1) and thus play a pivotal role in the establishment of the pre-metastatic niche. Accumulation of MDSCs in the TME in experimental animal models as well as in patients with different types of cancer (e.g., lung, head and neck, renal, breast, and prostate cancer) has been associated with disease progression and poor prognosis [28,29].

## 3. Main Phenotypic and Functional Characteristics of MDSCs

During the past decade, both the definition and classification of MDSC populations have become increasingly complex. MDSCs in tumor-bearing mice are well defined and classically divided into M-MDSCs (CD11b^+^/Ly6C^+^) and G-MDSCs (CD11b^+^/Ly6G^+^). Comprehensive transcriptomic and cell profile analysis identified MDSCs as a major TME population in PTEN/Smad4 deficient background of PCa tumors [30,31]. Further, a recent study confirmed that G-MDSCs represent the prevalent MDSC subpopulation in PC3 tumor-bearing mice and anti-Gr-1 treatment resulted in a significant reduction of these cells in the peripheral blood and the TME.

However, as noted in other tumor types [32], G-MDSCs may not be the predominant MDSC subpopulation in human patients, thus adding in the complexity of establishing murine models that precisely reflect aspects of the clinical PCa. Indeed, in humans, these cells are less clearly defined. Classically, they are described as lineage cells that coexpress CD11b and CD33 but lack HLA-DR. They are divided into two main MDSC subsets: M-MDSCs (CD14^+^) and G-MDSCs (CD15^+^). Additional functional markers have also been attributed to MDSCs, such as Arg-1, ROS, iNOS, IDO, and CD124, which all mediate immunosuppression [33]. Efforts being mounted on the phenotypic characterization of the MDSCs have highlighted subpopulations that have clinical significance in PCa patients. Vuk-Pavlović et al. first described a CD14^+^HLA-DR^low/−^ cell population with immunosuppressive properties that is isolated in greater percentage from the blood of PCa patients compared to age-matched healthy controls. They also observed that the levels of these circulating cells were further increased in patients receiving hormonal treatment [34]. Subsequently, Chi et al. reported that the CD33^+^CD11b^+^HLA-DR^−^CD14^−^ subpopulation is the predominant among newly diagnosed treatment-naïve PCa patients [35]. In a combined approach of studying immunosuppressive cells (MDSCs and regulatory T cells (Tregs)) in patients with mCRPC, Idorn et al. reported that a circulating CD14^+^HLA-DR^low/neg^ M-MDSC subset was significantly increased in the peripheral blood of patients compared to healthy donors [36]. More recently, Mehra et al. undertook a more comprehensive characterization of MDSCs in 46 mCRPC patients. They identified four phenotypically distinct subpopulations that were increased in the circulation and are associated with resistance to treatment [37]. Finally, Wen et al. focused on a G-MDSC subpopulation, defined as CD11b^+^CD33^+^CD15^+^ cells, and showed that these cells were upregulated in the stroma of metastatic sites (bone and lymph nodes) compared to the stroma of primary tumors [38]. Based on these data, the absence of a widely accepted protocol for the phenotypic characterization of MDSCs in PCa is unambiguous, necessitating the vital need for the development of novel strategies to organize all these data into a clinically meaningful consensus. This will provide recommendations to guide scientists in this field of MDSC-related research and serve as foundation for fruitful research projects.

## 4. Mechanisms Underlying MDSC-Mediated Immunosuppression in PCa

In recent years, studies have been focused on delineating the mechanism by which MDSCs induce PCa progression. As mentioned above, expansion and activation of MDSCs in the TME leads to immunosuppression and cancer development by targeting different innate and adaptive immunity cells and functions. Activated MDSCs exert their immunosuppressive and protumorigenic properties primarily through direct cell-cell contact or release of cytokines and short-lived factors. The PTEN-null prostate cancer mouse model recapitulates the main genetic alterations and disease hallmarks and thus is used for detailed assessment of the mechanistic role of MDSCs [39]. Moreover, in this non-human model, due to *PTEN* loss, elevated numbers of Gr-1^+^CD11b^+^cells are detected in the TME compared to wild-type mice. These numbers can be additionally increased in a disease progression and age-related manner [30,39,40]. In this context, proinflammatory cytokine response genes *CSF-1* and *IL1b* induced in prostate epithelial cells cause tumor development via MDSC expansion that further leads to immune suppression [40]. This phenotype can be reversed by treatment with the selective CSF-1 receptor inhibitor GW2580, which reduces the MDSCs infiltration and subsequently alleviates the associated immunosuppressive phenotype [40]. The most important expression of IL-10 by these PTEN-null prostate MDSCs were found to be responsible for the suppression of DC and macrophage maturation [40]. The inflammatory cytokine IL-23 is produced by MDSCs and has been recently linked to CRPC development, since it induces the transcription of AR target genes through STAT3-RORγ pathway, leading to the proliferation of cancer cells and tumor progression [41].

An important transcription factor that plays a central role in the generation and function of MDSCs is STAT3 [42,43,44]. pSTAT3 is activated by various cytokines such as IL-6, IL-1b, IL-10, GM-CSF, and VEGF secreted mainly in the TME by tumor cells [45]. In an orthotopic PCa mouse model, pSTAT3 inhibition in MDSCs abrogated tumor growth and metastasis [46]. Hellsten et al. showed that GM-CSF secreted by PCa cells promoted MDSC generation via pSTAT3. In response, MDSCs secreted IL-6, IL-1b, and IL-10, which induced immunosuppression [47]. Additionally, by using a STAT3 inhibitor, IDO levels were significantly reduced [47]. IDO catalyzes the degradation of tryptophan, leading to accumulation of metabolites such as kynurenine [48,49], that suppress T cells via inhibition of their expansion and recruitment of T regs [50,51]. Toll-like receptor 9 (TLR9) signaling in PCa cells can additionally stimulate the STAT3 pathway in MDSCs [52]. The TLR9 signaling pathway can induce PCa cell propagation and self-renewal in AR-indifferent disease state [53,54]. In order to study the immunosuppressive function of TLR9-mediated MDSCs, Won et al. used two different PCa cell lines with inducible *tlr9* implanted in C57/BL6 mice. TLR LIF–mediated STAT3 signaling in PMN-MDSCs induced the expression of S100A8, S100A9, and C/EBPb that, in turn, augmented tolerogenic activity by reducing T cell stimulation [52].

Metabolite depletion, such as for L-arginine (L-Arg), by MDSCs is another biological process that can lead to immunosuppression. L-Arg is an important non-essential amino acid crucial for mammalian immune system function. Its depletion shortens T-cell antigen receptor CD3ζ- chain mRNA half-life, leading to T-cell dysfunction [55,56]. Moreover, hydrolysis of L-Arg leads to the production of NO and ROS [57]. In PCa murine models, Bozkus et al. showed that extracellular Arg1 is transported into MDSCs by the upregulation of cationic amino acid transporter 2 (Cat2). Depletion of CAT2 abrogated the suppressive function of MDSCs on T-cells via reduction of intracellular L-Arg levels and led to further decreased tumor growth [58]. Hossain and colleagues highlighted the role of a TLR9-STAT3-Arginase-1 pathway in the inhibition of autologous CD8^+^ T-cell proliferation and cytotoxic activity by a G-MDSC population (Lin^−^CD15^HI^CD33^LO^) that is expressed in PCa patients [42]. Other alterations in the metabolism of MDSCs, such as upregulation of glycolysis, are also implicated in the establishment of an immunosuppressive TME [59,60]. A recent study using the TRAMP C2 CRPC tumor model suggested that the AR signaling pathway regulates the ability of MDSCs to suppress adaptive immunity. Inhibition of AR signaling on MDSCs with enzalutamide suppressed mitochondrial respiration via MPC/AMPK signaling pathways. The subsequent induction of glycolysis led to increased VEGF and Arg1 expression in MDSCs that further increased their suppressive activity and promoted tumor growth [61].

Reactive nitrogen species (RNS) are chemical modifiers that induce nitration of chemokines and receptors of T cells, leading to reduced T-cell infiltration and impaired T-cell function in various cancers [62,63]. Nitration of proteins is a form of post translational modification that converts tyrosine (Tyr) to 3-nitrotyrosine (3-NT), which differ in their chemical properties, and thus leads to different signaling activity [64,65]. In a CRPC mouse model, Feng et al. showed that MDSCs nitrate the lymphocyte-specific protein tyrosine kinase (LCK) at Tyr394. As LCK is a tyrosine kinase of the T cell receptor signaling cascade, nitration inhibited T-cell activation, leading to reduced interleukin 2 (IL2) production and proliferation [66]. Nitration of LCK by MDSCs is possibly mediated via induction of eNOS and iNOS/NO pathway [62].

Neutrophil elastase (NE) is a protease encoded by the *ELANE* gene with an established protumorigenic role in various cancer types [67,68,69]. It has been shown in murine models that NE stimulates tumor growth, while deletion of the corresponding gene results in tumor mass reduction [68]. There is evidence that the tumor-promoting capacity of NE is linked to its immunosuppressive properties [70,71]. In a PΤΕΝ-null PCa mouse model, Lerman and colleagues found that NE activity was significantly upregulated and, most importantly, that this was accompanied by expansion of MDSCs [72]. Additionally, they showed that a specific subpopulation of MDSCs produced NE at the TME of PCa [72]. Thus, NE production may confer the immunosuppressive effect of MDSCs and, in part, drive PCa development and progression.

CXCR4 is a chemokine receptor type 4 (CXCR-4) that signals through its ligand stromal cell-derived factor 1α (SDF1α or CXCL12). Normally, it is expressed at low levels but can be found at high concentrations in tumor cells, where it is mainly linked to metastatic potential [73]. MDSCs produce high levels of ROS that increases CXR4 levels, which, in turn, attract MDCSs in the TME [74]. Interestingly, CXCR4 has been shown to exert immunosuppressive properties by suppressing CD8^+^ T cell activity, increasing Tregs and inducing polarization of TAMs [75,76,77]. In PCa, CXCR4 expression induces the development of bone metastasis [78] that can be reversed upon its inhibition [79].

There is evidence that prostaglandin (PG)E2 is another factor implicated in the induction of MDSCs and their suppressive functions in cancer patients [80]. Tomic and colleagues showed that PGE2 potentiated the GM-CSF/IL-6–dependent induction of M-MDSCs [81]. Additionally, PGE2 reduces the capacity of the generated M-MDSCs in vitro to produce proinflammatory cytokines upon activation but increase their capacity to produce IL-27, IL-33, and TGF-β. These alterations lead to the induction of different subsets of Tregs, favoring IL-10 production by CD4^+^FoxP3^−^ type 1 Treg, and subsequently lower capacity to induce TGF-β-producing FoxP3^+^ Tregs by increasing IDO-1 expression [81]. Collectively, PGE2 M-MDSCs were correlated with suppression of T-cell proliferation, expansion of alloreactive Th2 cells, and reduction of development of Th17 and cytotoxic T cells. In prostate cancer patients, PEG-2 is expressed in high levels. Its expression is correlated with COX-2 inhibition [82,83], induction of M-MDSCs, and subsequent suppression of cytotoxic T cells [84].

Lastly, it is worth mentioning the impact of metabolic pathways in regulating MDSC function. In the tumor microenvironment, due to lipid accumulation, MDSCs undergo metabolic reprogramming and induce fatty acid uptake [85]. Fatty acid oxidation (FAO) as a source of energy for MDSCs in mice increases their immunosuppressive activity, while treatment with FAO inhibitors is able to improve anti-tumor immunity [86]. b-adrenergic receptor (b-AR), a stress signaling receptor, is also implicated in anti-tumor immune response. b2-AR is activated in MDSCs upon stress and its signaling enhances MDSCs accumulation in the tumor microenvironment [87]. Most important mechanistically, b2-AR signaling in MDSCs increases FAO among with the expression of the fatty acid transporter CPT1A, supporting the FAO mediated immunosuppression. Additionally, b2-AR signaling increases autophagy and activates the arachidonic acid cycle, which, in turn, induce the release of PGE2 [88]. Adrenergic signaling in prostate cancer cells affect apoptosis, angiogenesis, epithelial-mesenchymal transition, migration, and metastasis. Additionally, epidemiologic studies have shown that the blocking of β-adrenergic receptors results in reduced prostate cancer mortality [89]. Given the fact that, in mice with prostate cancer, b2-AR activation by stress induces MDSCs accumulation on tumor site, is reasonable to hypothesize that the immunosuppressive functions of such metabolic pathways in prostate cancer are correlated with MDSCs.

Irrespective of the underlying mechanism of action, it has been demonstrated that MDSCs are implicated in resistance to ICB and that their suppression reverses the observed resistance in PCa tumor models [13]. However, thus far, these promising murine data failed to be translated to and validated in humans. By exploiting existing expertise in studying the immune system, in vitro and in vivo efforts have been increased to bridge these knowledge gaps and revolutionize MDSC definition, identification, and ultimately, targeting of their immunosuppressive activity.

## 5. Clinical Significance of MDSCs in PCa

Several reports indicate a significant accumulation of various MDSC subtypes in the peripheral blood of PCa patients compared with healthy individuals. Their absolute numbers at baseline, as well as changes observed during different treatment interventions, could potentially represent a novel predictive/prognostic biomarker. A study including 23 PCa patients revealed that the CD14^+^HLA-DR^−^subset was increased compared to healthy donors and was significantly reduced after prostatectomy [90]. The frequency and the absolute numbers of MDSCs have been found to be significantly higher at the time of diagnosis of PCa patients compared to healthy age-matched donors [91]. Most importantly, MDSC levels have been linked with disease burden and levels of PSA in series of PCa patients. In addition, their levels have been correlated with clinical outcome upon different therapeutic interventions. In one of the first clinical studies investigating the significance of MDSCs as a biomarker in PCa, researchers compared newly diagnosed untreated PCa patients with patients under standard adjuvant therapy. The percentage of CD14^+^HLA-DR^low/−^ fraction was higher in the previously treated patients compared to the untreated [34]. This monocytic fraction was associated with PSA levels and was able to suppress the autologous T-cell proliferation [34]. The suppression of effector T cells by monocytic MDSCs was also confirmed by Idorn and colleagues [36]. In this study, the induction of M-MDSCs was correlated with established negative PCa prognostic markers, including elevated levels of lactate dehydrogenase (LDH) and PSA [36]. Notably, high M-MDSC levels before treatment were associated with a shorter median overall survival (OS) [36]. Similarly, high pretreatment M-MDSC (CD14^+^HLA-DR^−^) levels were linked with reduced OS in 28 CRPC patients treated with the GVAX vaccine and ipilimumab [92]. Koga et al. performed a phase II trial with 70 CRPC chemotherapy-resistant patients that were randomized to receive personalized peptide vaccination (PPV) plus herbal medicines (HM) or PPV alone. In contrast to the combination treatment, PPV alone resulted in an increase in M-MDSs. Patients with increased M-MDSCs after treatment demonstrated a significantly shorter survival compared to other patients [93]. Finally, a study investigating the addition of low dose cyclophosphamide to PPV in a cohort of 70 patients with mCRPC linked a decrease in the levels of M-MDSCs during treatment with an improvement in OS [94]. Taken together, these data indicate that the monocytic fraction of immunosuppressive MDSCs is linked with a higher disease burden and subsequent poor clinical outcomes in PCa. However, other studies pinpoint the granulocytic fraction as the main influencer of prognosis [35,42,95].

A study by Chi and colleagues reported that G-MDSCs were the predominant subset in PCa. This subpopulation was correlated with elevated serum IL-8 and IL-6 levels, as well as with reduced OS and poor prognosis, highlighting their prognostic significance in PCa [35]. Similarly, Hossain et al. demonstrated a correlation between cancer stage and G-MDSC levels [42]. SIeminnska and colleagues revealed higher percentages of G-MDSCs in the peripheral blood of PCa patients than in healthy donors, as well as lower percentages in treated patients compared to the untreated group. However, in this study, the level of G-MDSCs (or even M-MDSCs) was not correlated to tumor grade and clinical stage of the disease [95], most likely due to the limited number of patients included in the different subgroups. Thus, although PSA levels were correlated with MDSC levels, no correlation between PSA levels and clinical stage or tumor grade was detected. As PSA is a well-established and validated surrogate marker for disease stage, this finding indicates that, in this study, subgroup analysis cannot support definitive conclusions. Conclusively, more than 10 years after the pivotal studies on the prognostic value of MDSCs in PCa, there is still controversy regarding the most clinically relevant MDSC subpopulations. It is evident that more homogeneous clinical studies with large patient samples are needed to clearly define the main MDSC phenotype with predictive/prognostic value.

## 6. Limitations in the Study of MDSCs in PCa

Indeed, based on experimental findings, the investigation of the role of MDSCs in PCa progression and resistance to therapy is among the chief research priorities to improve the efficacy of ICB [30,37]. However, knowledge gaps have limited our ability to translate preclinical observations clinically. Significant causes of these gaps are PCa models that cannot reflect the complexity of human PCa with reasonable fidelity, differences between human and murine immune system, and challenges in profiling the immune status clinically. Furthermore, technical issues may account for a specific bias in the study of MDSCs, which should also should be considered. For example, as cryo-preservation can affect not only MDSC phenotyping but also functional assays [96], analysis should be performed on fresh samples (within 4 h of blood sampling). Moreover, obtaining a sufficient number of MDSCs from peripheral blood and tissues, especially at a purity that is suitable for functional analysis, can be a difficult task, especially considering the diversity of PCa patients included in the above-mentioned studies.

From a biological point of view, the plasticity of this heterogeneous cell population may add to the complexity of studying them. As several mediators that account for the MDSC function are subject to post-transcriptional regulation, genome, and RNA sequencing might be insufficient to accurately characterize MDSCs. In this case, other profiling routes, such as proteomics technology, should be considered. Furthermore, PCa is a complex disease that encompasses a range of advanced disease states with different predominant molecular biology and drivers of progression [97,98]. Therefore, study of MDSCs should consider the inherent biological heterogeneity of each disease state that may lead to significant differences in the clinical significance of various MDSCs subpopulations, influenced by their different biologic backgrounds. For example, increased frequency of both G- and M-MDSCs and increased levels of G-MDSCs have been reported in the peripheral blood of CRPC patients with MYC-amplified tumors and tumors with RB-1 loss, respectively [37]. More importantly, there are clear discrepancies observed in these studies as they used different markers to detect MDSCs among patients, which makes the results less generalizable. Consequently, it is mandatory that future studies should enroll more homogeneous groups of patients in terms of disease state (hormone naïve versus castration resistant), disease burden, and therapeutic regimens used in order to achieve a more precise definition of MDSCs and to establish a “universal language” in MDSC-related PCa research.

## 7. Targeting MDSCs

The development of novel insights into the genetic and molecular mechanisms that govern the immunosuppressive function of MDSCs has revitalized the concept of targeting these cells in order to enhance anti-tumor immune responses and to improve therapeutic interventions. Current strategies proposed to target MDSCs can be grouped into the following overlapping concepts: (1) deplete MDSCs; (2) impair MDSC function through inhibition of their immunosuppressive mediators; (3) disrupt MDSC recruitment and tumor trafficking; (4) stimulate MDSC maturation by promoting their differentiation. These are summarized in Table 1.

### 7.1. Depletion of MDSCs

Conventional anticancer therapies have been tested with conflicting results. There are limited and often inconsistent preclinical and clinical data regarding the effect of several chemotherapy agents on the expression of specific MDSC subpopulations with prognostic significance in various tumor types. Despite encouraging results in preclinical models with paclitaxel [106] and cyclophosphamide [107], they were not confirmed in a cohort of breast cancer patients [28]. Similarly, 5-fluorouracil (5-FU) was proven to induce apoptosis of MDSCs and enhance the antitumor immune response in tumor-bearing mice [108]. However, capecitabine, an oral prodrug of 5-FU, failed to consistently exert an inhibitory effect on a specific CD11b (+) MDSC subpopulation in patients with advanced pancreatic cancer [109,125]. Moreover, while FOLFIRI (5-FU with irinotecan) treatment was linked with an increase of MDSC percentage in the peripheral blood of patients with colorectal cancer, treatment with FOLFOX (5-FU plus oxaliplatin) was associated with a statistically significant decrease in the levels of circulating MDSCs [109]. Hence, their transient depletion induced by chemotherapy is currently considered a less promising strategy for the selective eradication of MDSCs, as these agents display a differential effect on the different MDSC subpopulations across various tumor types.

In addition, similar discrepancies have been reported for the use of targeted treatment. Various tyrosine kinase inhibitors (TKIs), including sunitinib [100], dasatinib [101], nilotinib, and sorafenib [102] have demonstrated a detrimental effect on the survival of MDSCs in vitro. This link was not as straightforward as expected in vivo. However, the addition of bevacizumab, a monoclonal antibody targeting VEGF-A, to conventional chemotherapy was associated with a significant decrease in the levels of circulating G-MDSCs in patients with metastatic non-small cell lung cancer [22]. Hence, as induction of hypoxia has been shown to promote recruitment of MDSCs in the tumor microenvironment [104], bevacizumab may increase the percentage of tumor-infiltrating MDSCs, as opposed to the reported reduction in the periphery. Nonetheless, encouraging results were reported recently when combinatorial strategies were deployed in a mCRPC mouse model by Lu et al. [103]. Although cabozantinib and BEZ235 (a PI3K/mTOR inhibitor) as single agents failed to exert any inhibitory effect on MDSCs, the combination of these drugs with immune checkpoint inhibitors managed to elicit robust antitumor immune responses and phase III clinical trials (e.g., CONTACT-02) are currently underway.

Finally, monoclonal antibody (moAb)-mediated MDSCs depletion has recently shown some promise. Qin et al. generated a novel peptibody (Pep-H6) targeting S100A9 on MDSCs surface that was shown to successfully deplete both G- and M-MDSCs and inhibit tumor growth in tumor-bearing mice [99]. Interestingly, following encouraging results from a phase II trial, tasquinimod, an oral S100A9 inhibitor, was tested in a large phase III trial that enrolled patients with mCRPC with bone metastases. Although there was a clear benefit for PFS, tasquinimod as monotherapy failed to improve OS. However, clinical trials aiming to exploit the synergistic effect of S100A9 inhibition with immune checkpoint blockade (ICB) are ongoing. Notably, gemtuzumab ozogamicin, a novel anti-CD33 antibody drug conjugate, has emerged as a propitious treatment strategy to effectively deplete MDSCs in various tumor types and augment anti-tumor immune responses in a clinically relevant manner [105]. As CD33 is a common surface marker for both G- and M-MDSCs, it is hypothesized that gemtuzumab ozogamicin could be used as part of combinatorial strategies to potentiate immune checkpoint treatment by converting “immune cold” tumors into “hot”. Clinical phase II trials in patients with solid tumors are being planned.

### 7.2. Impairment of MDSCs Function

Approaches aiming to neutralize the key mediators of MDSC immunosuppressive function have been also tested. Small molecule inhibitors targeting IDO, ARG, iNOS, cyclooxygenase 2 (COX2), STAT3, and phosphodiesterase type 5 (PDE5) have shown some preclinical activity, but clinical application of these agents as monotherapy has failed to improve the clinical outcome of patients with solid tumors. As an example, epacadostat, an IDO1 inhibitor, has shown limited efficacy as single agent in a phase II trial [126]. Moreover, entinostat, an oral histone deacetylase (HDAC) inhibitor that inhibits PCa tumor growth in vitro and in vivo in mouse models, has failed to demonstrate efficacy in a small cohort of mCRPC patients [112]. However, combination strategies have attempted to exploit the synergistic effects of these molecules with chemotherapy or immunomodulatory agents and produced interesting results that warrant further investigation. In particular, ASP9853, an iNOS inhibitor, in combination with a taxane appeared to be active in patients with various advanced solid tumors. Nevertheless, concerns raised due to toxicity issues halted any further development for this strategy [110]. On the other hand, ICB-based combinations seem to provide a better tolerated option and many clinical trials are underway. Hence, following promising results in PCa preclinical models [66], reactive nitrogen species (RNS) are being combined with ICB for the treatment of patients with CRPC. Similarly, a phase I/II clinical trial that is currently accruing patients is testing the addition of epacadostat to a brachyury-targeted antitumor vaccine, an IL-15 superagonist, and a TGF-β TRAP/anti-PD-L1 antibody in patients with mCRPC [111]. Encouraging preliminary results have also been reported from a phase Ib trial investigating the addition of vorinostat, a HDAC inhibitor, to pembrolizumab in a small cohort of patients with PCa [113]. Furthermore, inhibition of arginase has been shown efficacious combined with anti-PD1 blockade in vitro and in tumor-bearing mice; a phase 2 clinical trial has been initiated to clinically test this combination strategy [114]. Finally, Hossain et al. have developed a novel strategy to target the immunosuppressive function of MDSCs by TLR9-targeted STAT3 inhibition through si-RNA technologies in patients with CRPC experiencing disease progression [42]. These results have provided a solid background from which to launch a clinical trial.

### 7.3. Hampering MDSCs Recruitment

Inhibition of molecular pathways that regulate MDSC tumor infiltration has also been investigated in various tumor types. Indeed, treatment of tumor-bearing mice with a CCR-5 or a CSFR1-targeted tyrosine kinase antagonist improved the efficacy of anti-PD1 treatment and impeded tumor growth by significantly decreasing the percentage of intra-tumoral MDSCs [115,116]. In fact, previous reports implicating these pathways in the emergence of immune-mediated resistance to androgen blockade [117] also suggest that CSFR1 and CCR-5 may be promising targets for augmenting the efficacy of immunotherapy approaches in patients with CRPC. Additionally, inhibition of CXCR2 signaling appears to directly decrease the accumulation of tumor infiltrating MDSCs in PCa models. Wang et al. reported inhibition of tumor growth and improved survival outcome with the addition of a CXCL2 antagonist SB255002 to ICB in the PTEN^pc−/−^Smad4^pc−/−^ PCa model [30]. Moreover, a pivotal work by Lopez-Bujanda et al. has demonstrated increased efficacy of anti-CTLA4 blockade with the addition of anti-CXCR2 (IL-8 receptor) targeted treatment in the MyC-CaP mouse model. However, this beneficial effect appears to be heavily dependent on the context and the timing of therapy administration. As studies observed an IL-8 upregulation following progression on ADT, enhanced anti-tumor responses might be induced with initial treatment with ADT in the hormone sensitive disease and not after the emergence of CRPC [118]. A phase Ib/II clinical trial is currently underway testing this combination in patients with castration-sensitive PCa (NCT03689699).

### 7.4. Targeting MDSC Development/Maturation

Impairment of immature myeloid cell (IMC) differentiation into immune effector cells is linked with the accumulation of MDSCs and their immunosuppressive capability. Therefore, strategies to promote their differentiation process or even to transform them into immunostimulatory cells may be beneficial. Although convincing clinical data are missing, current research in this field holds promise for the development of future therapeutic combinations. Indeed, a growing amount of evidence suggests that various agents, including chemotherapy, differentiation-promoting vitamins, inhibitors of the metabolic stress axis, and facilitators of normal myelopoiesis could induce MDSC maturation. Vitamin D3 [119], as well as vitamin A and all-trans retinoic acid (ATRA) [120] have been reported to promote differentiation of MDSCs into macrophages, dendritic cells, and granulocytes in murine models and human studies. Notably, ATRA treatment has been shown to increase the efficacy of ICB [127] and antiangiogenic treatment [128] in melanoma patients and breast cancer models, respectively. Furthermore, preclinical data suggest that casein kinase 2 inhibitors can effectively modulate MDSCs in the TME by disrupting aberrant myelopoiesis in tumor-bearing mice. Interestingly, combination treatment with anti-CTLA4 further enhanced the anti-tumor effect and significantly inhibited tumor growth compared with ICB alone in various tumor models [124]. Similarly, genetically provoked upregulation of IRF8 expression, a transcriptional activator of PMN-MDSC maturation, decreased the percentage of MDSCs in the TME by promoting cellular lineage differentiation. Encouragingly, anti-CTLA4 treatment significantly decreased tumor growth in IRF8-transgenic mice compared to wild-type mice [129]. Finally, targeting endoplasmic reticulum (ER) or metabolic stress has also shown some potential [130]. To this end, pivotal studies have assessed the role of chemical chaperones such as tauroursodeoxycholic acid (TUDCA) [121], inhibition of AMPKa [122] or PERK [123], and etomoxir, an irreversible CPT1a inhibitor that suppresses fatty acid oxidation [59] in transforming MDSCs into immune-enhancing cells. Redirection of MDSC function has been shown to be feasible. This strategy has gained scientific interest, and research funding has been directed towards drug development.

While these data provide compelling evidence that this approach could represent a novel MDSC-directed treatment strategy, the mechanistic translational studies that link the preclinical observation to the clinical outcome of the patients are lacking. However, the number of patients included in these studies is small and further confirmation in larger studies is certainly needed.

## 8. Future Perspectives

Despite the limitations, the above-presented data support the notion that MDSCs play a key role in the immune-resistant phenotype in PCa. A better understanding of the role of MDSCs in PCa progression and resistance to therapy will provide an insight into the clinical relevance of MDSCs. To better recognize their role and elucidate their contribution to PCa growth and progression, it is necessary to accurately characterize the subsets of MDSCs in PCa patients and gain a better understanding of their generation, expansion, and function in the peripheral blood as well as the TME. Because of their heterogeneous composition, phenotyping these cells with fidelity requires a multicolor approach (CyTOF) with the simultaneous use of different markers so that the expansion of all MDSC subsets can be appreciated. In addition, the confirmation of the immunosuppressive capacity—a hallmark of their activity—of the identified MDSC subpopulations is considered mandatory. In fact, as these cells share the same functional properties in humans and mice, the development of functional assays will allow researchers to link MDSCs in murine models to clinical observations in patients and thus bridge the gap between bench and bedside. This process of precise definition of MDSCs through identification of specific phenotypic markers combined with the utilization of functional characterization will provide the foundational observations for future MDSC-related applied research.

The second step would be to measure the changes in MDSC frequency and functionality through the course of the disease. This study would deliver robust information on the prognostic and predictive significance of MDSCs in the various disease states of PCa (hormone-naïve, castration resistant, AR-indifferent), and ultimately, it will inform the development of MDSC-targeting therapies.

Ultimately, improving the understanding of MDSCs will be necessary to efficiently develop strategies to monitor and then target tumor-related immunosuppression. Thus far, the efficacy of strategies combining molecular targeted therapies with immunotherapy has shown promise, but these clinical observations need confirmation and the development of predictive biomarkers that will lead to a truly targeted application. Future projects will need to generate high-quality datasets, which will enrich our knowledge of MDSCs and generate the critical knowledge that will lead to the development of MDSC-targeting combinatorial strategies in order to increase the efficacy of ICB in PCa.

## Figures and Tables

**Figure 1 cells-11-00020-f001:**
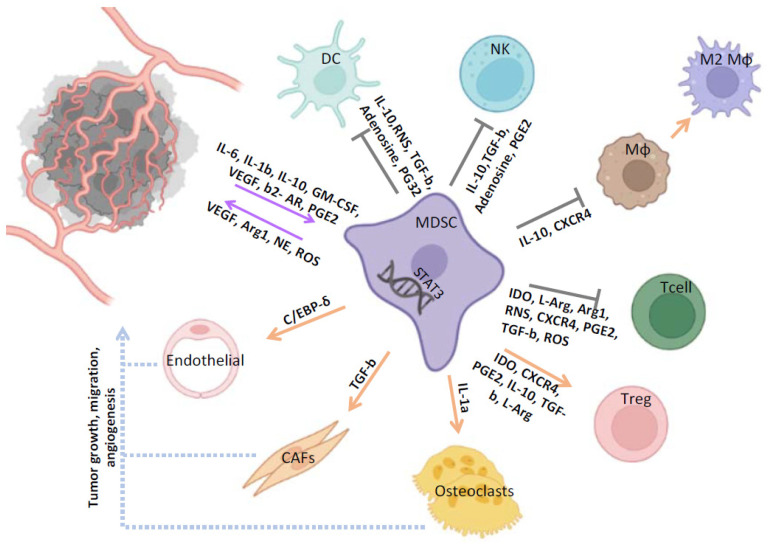
The role of MDSCs in the tumor-associated microenvironment of PCa. MDSCs exert their tumor promoting and immunosuppressive functions through induction of different signaling pathways and release of various factors. MDSCs suppress NK, Mφ, DC, and the cytotoxic activity of T cells, and also induce Tregs and M2 immunosuppressive phenotypes. MDSCs also participate in tumor formation, metastasis, and migration through interaction with CAFs, endothelial cells, and osteoclasts. MDSC, myeloid-derived suppressor cell; DC, dendritic cell; NK, natural killer; MΦ, macrophages; T cell, T lymphocytes; Treg, T regulatory cell; CAF, cancer-associated fibroblast.

**Table 1 cells-11-00020-t001:** Summary of current strategies proposed to target MDSC-mediated immunosuppression.

Method of Action	Target	Agent	Cancer Type	Reference
Depletion of MDSCs	S100A9	Tasquinimod	Advanced cancer	[99]
	Tyrosine Kinases Inhibitors	Sunitinib, nilotinib, dasatinib, sorafenib, cabozantinib plus BEZ235	Prostate cancer	[100,101,102,103]
	VEGF	Bevacizumab	NSCLC	[22,104]
	CD33	Gemtuzumab ozogamicin	Solid tumors, lymphoma, sarcoma	[105]
	Monotherapy chemotherapy	5-FU, Paclitaxel, Cyclophosphamide	Cancer	[106,107,108]
	Combination chemotherapy	5-FU/Oxaliplatin	Colorectal cancer	[109]
Impairment of MDSC function	iNOS	ASP9853/taxane	Advanced cancer	[110]
	LCK/PD1/CTLA4	RNS/ICB	Prostate cancer	[66]
	IDO1, IL15, PDL1, TGF-β TRAP	Epacadostat/brachyury-targeted antitumor vaccine	CRPC	[111]
	HDAC, HDAC/PD1	Entinostat, vorinostat/pembrolizumab	Prostate cancer	[112,113]
	ARG/PD1	INCB001158	Colorectal cancer	[114]
	STAT3	siRNA	CRPC	[42]
Blocking MDSC recruitment	CCL5/CCR5	Maraviroc	TNBC	[115]
	CSFR1	PLX3397	Pancreatic cancer, prostate cancer	[116,117]
	CXCL2	SB255002	Prostate cancer	[30]
	CXCR2	BMS-986253	Prostate cancer	[118]
Promotion of MDSC differentiation		Vitamin D3, Vitamin A, ATRA	Head and neck carcinoma, colon cancer, breast cancer, melanoma	[119,120]
	UPR	TUDCA	cancer	[121]
	AMPKa	Metformin or Aica-R	Lung cancer, ovarian cancer, thymoma, melanoma	[122]
	PERK	AMG-44, GSK-2606414	Cancer	[123]
	CPT1a	Etomoxir	Lung cancer, colon cancer	[59]
	Casein kinase 2	BMS-595, BMS-699, BMS-211	Lung cancer, breast cancer, colon cancer, lymphoma	[124]

## Data Availability

The data presented in this study are available on request from the corresponding author.

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
