# Peer review of "Myeloid-Derived Suppressor Cells in Prostate Cancer: Present Knowledge and Future Perspectives"

_cells, 2021, doi:10.3390/cells11010020_

Round 1
Reviewer 1 Report
The authors summarized the research progresses about MDSC inprostate cancer and explored the potential significance in cancer therapy. On the whole, the literature review is comprehensive, which can be helpful to readers in this field.
Major concers:
- The role and therapeutic potential of MDSC in cancer have been recognized in the field. Whether the author puts forward some new ideas on the particularity and treatment of MDSC in prostate cancer.
- The author has more comprehensive current clinical research data. It would be better if he could more include the latest research results of MDSC, especially the important research results of mechanism.
minor concerns:
- The figures looks not so good. It could be improved.
Author Response
Response to Comments from Reviewer 1
Major concers:
Comment 1:The role and therapeutic potential of MDSC in cancer have been recognized in the field. Whether the author puts forward some new ideas on the particularity and treatment of MDSC in prostate cancer.
Response 1:
We appreciate your insightful comment. We believe that the present review makes a valuable contribution to the field firstly by summarizing all the potential limitations in the studies of MDSCs particularly in prostate cancer (“Limitations in the study of MDSCs in PCa”). Furthermore, by providing an insight into our future projects and methods that we are currently deploying to address these issues (functional assays, CyTOF, etc) in the “Future perspectives” section, we put forward a multi-step process that will enrich our knowledge in MDSCs identification, functional characterization and ultimately therapeutic targeting.
Comment 2: The author has more comprehensive current clinical research data. It would be better if he could more include the latest research results of MDSC, especially the important research results of mechanism.
Response 2:
Thank you for this valuable suggestion. In the revised version of this manuscript the latest research results regarding the mechanism have been included. These include the role of Prostaglandin (PG)E2 in suppression of cytotoxic T cells and in induction of Tregs. Moreover, the impact of metabolic pathways such as Fatty acid oxidation and b-adrenergic receptor signaling in prostate cancer are analyzed in the section Mechanisms underlying MDSC-mediated immunosuppression in PCa
minor concerns:
Comment 1:The figures looks not so good. It could be improved.
Response 1: Thank you for this helpful suggestion. In the revised version of this manuscript the figures have been improved. In figure 1 we included the factors that contribute to immunosuppressive and tumor promoting functions of MDSCs in PCa.
Reviewer 2 Report
In this paper Koninis et al., have nicely summarized the role of MDSCs in prostate cancer and properly outlined different ways of targeting MDSCs in prostate cancer. But there are major concerns regarding different sections of the paper.
1- Figure 1 is incomplete. In this figure, authors should show in the graph what are the know immunosuppressive pathways or molecules involve in MDSC mediated suppression of each immune and non-immune cells listed in the picture. For example, what are the know mechanisms of how MDSCs suppress T cells? This info needs to be added in the figure.
2- There are a lot of new published data about MDSCs and the mechanisms of their immunosuppressive function such as release of PGE2 by MDSCs and the role of adrenergic signaling in MDSC immunosuppressive function and metabolism. Adrenergic signaling is a relevant pathway in resistant prostate cancer. It will increase the strength of the paper if these new data about MDSCs added in the paper
3- It will improve the paper if authors summarize different ways of targeting MDSCs in a table.
Author Response
Response to Comments from Reviewer 2
Major concers:
Comment 1: Figure 1 is incomplete. In this figure, authors should show in the graph what are the know immunosuppressive pathways or molecules involve in MDSC mediated suppression of each immune and non-immune cells listed in the picture. For example, what are the know mechanisms of how MDSCs suppress T cells? This info needs to be added in the figure.
Response 1:
Thank you for the insightful comment. In the revised version of this manuscript figure 1 has been improved. The known factors that contribute to immunosuppression and tumor promoting properties of MDSCs in PCa have been included.
Comment 2:There are a lot of new published data about MDSCs and the mechanisms of their immunosuppressive function such as release of PGE2 by MDSCs and the role of adrenergic signaling in MDSC immunosuppressive function and metabolism. Adrenergic signaling is a relevant pathway in resistant prostate cancer. It will increase the strength of the paper if these new data about MDSCs added in the paper
Response 2:
Thank you for this valuable suggestion. In the revised version of this manuscript the latest research results regarding the mechanism have been included. These include the role of Prostaglandin (PG)E2 in suppression of cytotoxic T cells and in induction of Tregs. Moreover, the impact of metabolic pathways such as Fatty acid oxidation and b-adrenergic receptor signaling in prostate cancer are analyzed in the section Mechanisms underlying MDSC-mediated immunosuppression in PCa
Comment 3:It will improve the paper if authors summarize different ways of targeting MDSCs in a table.
Response 1:
This is an excellent suggestion. In the revised version of this manuscript we have added Table 1 that summarizes current strategies being investigated to target MDSCs mediated immunosuppression.
Round 2
Reviewer 1 Report
No any other concerns about this review.
Reviewer 2 Report
Authors have revised the manuscript sufficiently. I have no more comments.